# Mental Health Status of French School-Aged Children’s Parents during the COVID-19 Lockdown and Its Associated Factors

**DOI:** 10.3390/ijerph191710999

**Published:** 2022-09-02

**Authors:** Stéphanie Bourion-Bédès, Hélène Rousseau, Martine Batt, Pascale Tarquinio, Romain Lebreuilly, Christine Sorsana, Karine Legrand, Rabah Machane, Cyril Tarquinio, Cédric Baumann

**Affiliations:** 1Centre Hospitalier de Versailles, Service Universitaire de Psychiatrie de l’Enfant et de l’Adolescent, 78157 Versailles, France; 2UR 4360 APEMAC (Health Adjustment, Measurement and Assessment, Interdisciplinary Approaches), University of Lorraine, 54000 Nancy, France; 3Methodology, Data Management and Statistics Unit, University Hospital of Nancy, 54000 Nancy, France; 4InterPsy, GRC Team, University of Lorraine, 54000 Nancy, France; 5Pierre Janet Center, University of Lorraine, 57000 Metz, France; 6Clinical Investigation Center, INSERM, University Hospital of Nancy, 54000 Nancy, France

**Keywords:** anxiety, COVID-19 pandemic, parents, resilience, family support

## Abstract

The COVID-19 pandemic has led to widespread social isolation. This study aimed to determine anxiety levels among parents of school-aged children and investigate the associated factors. Data on sociodemographic characteristics, living and working conditions, family relationships, social support (MSPSS) and health status (SF-12) were collected from French parents through an online survey. The Generalized Anxiety Disorder Scale (GAD-7) was used to assess anxiety. Logistic regression analysis was performed to identify the factors associated with moderate to severe anxiety. Among 698 parents, 19.2% experienced moderate to severe anxiety. A low level of resilience (OR = 4.3, 95% CI: 2.7–6.7) and confirmed COVID-19 cases involving hospitalization (OR = 3.8, 95% CI: 2.0–7.3) among individuals in one’s household or in the family circle were found to be the main risk factors for moderate to severe anxiety. Other factors were also identified: a level of education less than high school (OR = 2.1, 95% CI: 1.3–3.2), conflicts at home (OR = 2.3, 95% CI: 1.4–3.7), noises outside the home (OR = 2.0, 95% CI: 1.0–3.9), confirmed cases not involving hospitalization (OR = 1.8, 95% CI: 1.0–3.1) and suspected cases (OR = 1.9, 95% CI: 1.0–3.8). Family support was a protective factor. These findings suggest some need for support programs to help parents cope with public health crises and work-family challenges.

## 1. Introduction

Coronavirus disease 2019 (COVID-19) emerged in Wuhan, China, in December 2019 [1] and began to spread in France in January 2020. The World Health Organization (WHO) declared a pandemic status on March 11th, 2020, due to the extent and severity of the epidemic [2]. Many countries implemented containment measures to curb the spread of SARS-CoV-2, such as travel restrictions, social distancing, home lockdowns and school closures. Parents and children had to work and study at home [3] and experienced long periods of physical isolation from their usual environments. For people who stayed at home and were socially isolated to slow the spread of the virus, isolation also affected their mental health, causing stress-related symptoms, depression and anxiety [4] in children, adolescents and adults [5,6]. In addition, the limited living space during home lockdown affected mental health [7]. As school closures affected over 1.5 billion school-aged children globally during the lockdown [8], children and adolescents experienced a long period of isolation from their extended families, teachers and peers [9]. Parents staying at home faced hardships in taking care of their children’s living and learning conditions while balancing their personal lives, children’s education and work without any external help [10]. As reported in recent studies [11,12,13], long periods of social isolation during the pandemic are well known to be associated with a deteriorating family climate and more escalating conflicts at home [14].

Few studies have focused on the mental health of parents during the COVID-19 pandemic [3], although parents who had limited social interaction and remained at home with their children may have been particularly vulnerable during this period [15]. Thus, it seems important to monitor the prevalence of mental disorders to identify the impact of the pandemic on parental well-being. Among South Korean parents, symptoms of moderate to severe depression and mild depression were reported at a level of 17.5% and 29%, respectively, during COVID-19 school closures [16]. Among American parents of children under 18 years old, nearly half (46%) reported a high stress level [17]. In a nationwide study in a United Arab Emirate community, 71% of the adult population recruited from four schools reported anxiety, and 38% had moderate to severe anxiety as measured by the seven-item Generalized Anxiety Disorder Scale (GAD-7) [18]. In addition to the fear and stress of being infected by the new coronavirus, some studies reported that the parent–child relationship also affects the mental health of parents. In one online consultation during the COVID-19 pandemic, parents voiced many practical problems, such as how to get along with their children and how to deal with conflicts with their children and adolescents. It was also reported that many parents participated in network lectures to facilitate communication with their children, resolve the family’s parent–child conflicts and improve the quality of the parent–child relationship [3]. As parents’ mental health can further affect their children’s physical and mental health [3,19], it seems to be urgent to pay attention to the mental health of parents during the COVID-19 pandemic. Some families also experienced more financial strain as a result of increasing unemployment [15].

A recent systematic review of 87 studies was conducted to investigate the anxiety levels and common risk factors among different sample populations, including parents, in relation to the COVID-19 pandemic. Risk factors such as being female, having pre-existing mental conditions, having a low socioeconomic status, having increased exposure to infection and being young all contributed to worsened anxiety [20]. Social support, as a supportive resource obtained by individuals from others, was reported to be an important factor affecting individual mental health that can help parents cope with crises [21]. The results of previous studies have shown that perceived social support was negatively associated with psychological distress among parents [21]. Past studies of other pandemics have also shown that resilience plays a key role in mitigating the negative effects of stress on parents. Recent studies further point in this direction in the context of the COVID-19 pandemic [22]. Although data on youths are starting to be published, relevant data from France on this topic are lacking.

Thus, this study aimed to:(a)Examine the prevalence of anxiety among school-aged children’s parents under lockdown in the Grand Est region, which was the first French region affected by the disease during the first wave of the pandemic. We expected to observe higher levels of psychological well-being in parents compared to French young adults during the same period.(b)Identify whether sociodemographic characteristics, living and working conditions, family relationships, the availability of parents to assist with their children’s homework, social support and parent health status characteristics were associated with anxiety among school-aged children’s parents during the national part-time lockdown due to the COVID-19 outbreak to develop interventions to provide parents with suitable guidance and assistance to prepare for future waves of the pandemic or comparable events. The initial hypothesis was that parents’ anxiety levels were likely to vary according to individual (gender, resilience) and probably socio-environmental (lack of social support) parameters.

## 2. Methods

### 2.1. Procedure

This was a cross-sectional analysis of data from the observational Feelings and Psychological Impact of the COVID-19 Epidemic among Children and Adolescents in the Grand Est Area (PIMS2-CoV19) study. To conduct this study, schools were randomly selected from a complete list of middle and high schools in the Academy of Metz-Nancy using proportionate stratification for the baseline school identification and recruitment; then, an invitation letter and the online survey were forwarded through school institutional mailing lists. The survey questionnaire consisted of three parts: sociodemographic data, living conditions and the measurement of perceived mental health status assessed by validated self-report questionnaires. The questionnaires took approximately 20 min to complete. Data collection took place from 26 May to 6 July 2020, while the country was under a partial lockdown. Online consent was obtained from all participants, and the study was conducted in full compliance with the principles of the Declaration of Helsinki. The study protocol was approved by the Commissioner for Data Protection (Comité National Informatique et Liberté-registration 2220408).

### 2.2. Participants

The inclusion criteria were (a) parents aged 18 years or older and (b) parents with at least one child aged 8–19 years living at home during the lockdown. Data were collected from 850 parents. Participants with an incomplete questionnaire were excluded from the analyses. Hence, the final sample consisted of 698 parents. No information was available for families who did not respond to the invitation letter. Sociodemographic data are shown in Table 1. The parents consisted of 600 females (86.0%) and 98 males (14.0%), and they were mostly 35 to 45 years of age (51.7%). Almost seventy-five percent of the participants were married or living with the other parent (74.5%). In total, 2066 children under the age of 19 were enumerated. Of these 2066 children, 670 were under 8 years old, 698 were between 8 and 11 years old and 698 were over 11 years old. The number of children living at home ranged from one to four. Over half of the parents had a level of education which was higher than high school (62.1%), and 86.5% of the participants were employed before the lockdown. Most of them indicated having full-time work (63.6%) before the pandemic, and 77.6% stated they had changed their work habits due to the COVID-19 outbreak and the lockdown, with 30% reporting interrupted work.

### 2.3. Measures

In addition to the questionnaire designed to investigate demographic variables, data on living characteristics were collected, and the following measures were employed.

#### 2.3.1. Living and Learning Characteristics

Parents were also asked about their living conditions, such as home location, access to an outdoor area and the presence of a relative or acquaintance infected with COVID-19. They were also asked whether there were family conflicts and noises inside or outside the residence during the pandemic. Information about the time spent by parents on children’s homework for school was also collected. All data were obtained at the time of the online survey.

#### 2.3.2. Anxiety

Anxiety was screened based on the GAD-7, which is a useful self-report anxiety scale used both in clinical practice and research [23] to assess symptom severity over the previous two weeks [24]. Example items include “Not being able to stop or control worrying”, “worrying too much about different things” and “becoming easily annoyed or irritable”. Each of the seven items is scored based on a 4-point Likert scale ranging from 0 (“not at all”) to 3 (“nearly every day”). Total scores range between 0 and 21, with higher scores indicating more severe anxiety symptoms. A score of 5–9 is considered “mild anxiety”, 10–14 is considered “moderate anxiety”, and 15–21 is considered “severe anxiety” [24]. The French version of the measure was used, as it demonstrated good internal consistency [25]. In this study, the Cronbach’s alpha coefficient was 0.910. 

#### 2.3.3. Health-Related Quality of Life

The French generic SF-12 instrument was used, as it provides a valid clinical assessment of self-reported quality of life [26]. It includes a subset of 12 items from the eight subscales of the SF-36: physical functioning, role-physical, bodily pain, general health, vitality, social functioning, role-emotional and mental health. Scores from the two summary measures that comprise the SF-12 are calculated: a mental health component summary (MCS) score and a physical health component summary (PCS) score. Total scores are transformed into standardized scores that range from 0–100, where higher scores indicate better self-reported quality of life.

#### 2.3.4. Self-Perceived Social Support

The Multidimensional Scale of Perceived Social Support (MSPSS) was used to measure self-perceived social support. This scale comprises 12 items assessing social support from three sources: family, with example items such as “My family tries to help me” and “I can talk about my problems with my family”; friends, with example items such as “I can count on my friends when things go wrong” and “I have friends with whom I can share my joys and sorrows”; and significant others [27]. Each item is rated on a 7-point Likert scale ranging from 1 (“very strongly disagree”) to 7 (“very strongly agree”). The item scores are summed for each dimension following the instructions for scoring the MSPSS: the score of each dimension ranges from 1 to 7, with higher scores indicating higher perceived social support. The French version of the scale has been previously confirmed to have good internal reliability and reproducibility [28]. In this study, the Cronbach’s alpha coefficients for the friends, family and other dimensions were 0.956, 0.926 and 0.904, respectively.

#### 2.3.5. Resilience

The ability to recover from stress was assessed using the Brief Resilience Scale (BRS) [29]. The scale is composed of six items (e.g., “I tend to bounce back quickly after hard times” and “I usually get through difficult times with little trouble”), of which items 2, 4 and 6 are negatively phrased. Each item is rated on a 5-point Likert scale ranging from 1 (“very strongly disagree”) to 5 (“very strongly agree”), and the BRS is scored by reverse coding items 2, 4 and 6 and then calculating the mean, which ranges from 1 to 5. A higher score indicates greater resilience. Scores ranging from 3 to 4.3 are considered to indicate a normal level of resilience. A score < 3 is indicative of low resilience, whereas a score > 4.3 indicates high resilience [30]. The French version of the scale has demonstrated satisfactory psychometric properties [31]. In this study, the Cronbach’s alpha coefficient was 0.750.

### 2.4. Statistical Analysis

First, descriptive analysis was carried out to describe the living conditions and mental health status of the parents. Continuous variables are described by the mean and standard deviation or the median, as appropriate, and categorical variables are described by percentages. Second, a logistic regression analysis was performed to explore the association of each independent variable with the outcome i.e., moderate to high levels of anxiety. Thus, the modeled probability was a GAD-7 score higher than 10 [24]. The influence of sociodemographic characteristics, working conditions, living conditions, family conflicts, time spent on children’s homework, concerns regarding the health threat posed by COVID-19, parents’ health-related quality of life physical domain scores, resilience and self-perceived social support scores were investigated. The PCS was analyzed in binary form, using the median to split the sample, and resilience was also analyzed in binary form because of the recommended cutoff value of <3 for low resilience [30]. Variables with *p* values < 0.1 in the bivariate analysis were included in the multivariate analysis. Odds ratios (ORs) and 95% confidence intervals (CIs) were estimated. Statistical significance of independent variables in the multivariate model was considered at *p* value < 0.05. The goodness of fit was assessed by calculating the model determination coefficient (R2) and the percentage that was predicted to be correct by the model. The lack of multicollinearity was verified by calculation of variance inflation index. The Hosmer and Lemeshow test allowed the comparison and selection of the best multivariable model. Analyses were performed using SAS 9.4 (SAS Inst., Cary, NC, USA).

## 3. Results

### 3.1. Living Condition Characteristics

Living condition characteristics are shown in Table 2. More than half of the participants lived in rural areas (60.7%), and 5.5% reported having no access to outdoor areas. Of the 698 parents, 13.2% reported never being out of the house during the lockdown, and 30.7% reported being out of the house once or less than once a week. Parental reports on their children’s online learning attitudes showed that 36.8% of the parents spent between 2 and 4 h per day helping their children, and 43.1% reported spending 2 h or less per day. Conflicts with family members who resided in the same location during the lockdown were noted by 18.6% of the sample, and difficulties isolating at home were reported by 15.3% of the sample. Less than one-third of the parents (30.1%) stated that someone at home or a relative or acquaintance had been infected with COVID-19.

### 3.2. Parental Mental Health and Social Support

Table 3 provides the means and standard deviations of the scores for anxiety, health-related quality of life, social support and resilience in the total sample. Of the 698 parents, 48.6% had no symptoms of anxiety, whereas the proportions of parents with mild, moderate, and severe anxiety were 32.2%, 13%, and 6.2%, respectively. The mean PCS score was 69.6 (SD = 13.0) (median 74.2, interquartile range 64.7–77.8), and the mean MCS score was 55.4 (SD = 16.9) (median 58.4, interquartile range 42.4–69.2). The mean MSPSS total score was 5.5 (SD = 1.2). The mean scores for support from family, friends and significant others were 5.5 (SD = 1.3), 5.3 (SD = 1.4) and 5.7 (SD = 1.3), respectively. The mean resilience score for the overall sample was 3.5 (SD = 0.8), and 155 parents (22.2%) showed a low level of resilience (BRS < 3).

### 3.3. Factors Associated with Moderate to Severe Parental Anxiety during the Lockdown

Table 4 shows the results of the bivariate and multivariate analyses. The model determination coefficient (R^2^) was 0.3. VIF varied from 1.01682 for “parent education level” to 1.17464 for “BRS family sub-score”, indicating the lack of collinearity. Predictors that were significantly associated with moderate to severe anxiety included some sociodemographic, environmental and psychological factors. Specifically, those who reported tension and conflicts at home (OR = 2.3, 95% CI: 1.4–3.7), noises outside the residence (OR = 2.0, 95% CI: 1.0–3.9) and confirmed COVID-19 cases among individuals in the home or in the family circle, whether these cases were confirmed and involved hospitalization (OR = 3.8, 95% CI: 2.0–7.3), confirmed and did not involve hospitalization (OR = 1.8, 95% CI: 1.0–3.1) or suspected (OR = 1.9, 95% CI: 1.0–3.8), were associated with a moderate to severe level of anxiety. In terms of psychological factors, parents with high levels of family support (OR = 0.7, 95% CI: 0.6–0.8) were less likely to score at moderate to severe levels for anxiety. Those with a low level of resilience were more likely to score above the cutoff for moderate anxiety (OR = 4.3, 95% CI: 2.7–6.7). No associations were obtained between support from others and friends. Regarding sociodemographic characteristics, those who reported less than a high-school education had a higher probability of anxiety symptoms (OR = 2.1, 95% CI: 1.3–3.2).

## 4. Discussion

Important to the primary aim of this study was the finding that a majority of the parents experienced anxiety symptoms during the lockdown due to the COVID-19 outbreak. This result highlights parents’ perception of COVID-19 as a much more serious threat than A/H1N1 influenza; the majority of the French general population did not consider A/H1N1 influenza as a serious threat during the A/H1N1 influenza pandemic [32]. The rate of moderate to severe anxiety among parents in our study was 19.2%, which was below the rate of moderate to severe anxiety among French young adults living in the same area (25%) [33]. Using the same cutoff points (GAD-7 scores ≥ 10), the rate of moderate to severe anxiety among our sample was lower than the rate of 38% among the adult population in the United Arab Emirates [18] and higher than the rate of 4% reported among Chinese parents of students from primary school to college in mid-March 2020 [3] and the rate of 6.7% reported among Swiss parents of children aged 12–17 years [34]. The high prevalence observed in our study might be explained by the use of an online assessment after a period of isolation that lasted for almost two months, with massive media coverage and daily reports of fatalities in the news media (i.e., 1555 hospitalizations in intensive care units (ICUs) and 18 195 deaths at the beginning of the survey) (www.santepubliquefrance.fr (accessed on 3 June 2020) [35]. Parents might also have been particularly distressed by work responsibilities and managing their children’s schooling while having no certainty regarding the pandemic’s end. Moreover, they might have been particularly distressed because the Grand Est region was one of the three French regions that was the most severely affected by COVID-19 during the first lockdown, with proximity to a greater number of individuals infected with COVID-19 potentially being an added stressor for our sample.

Our findings did not show significant gender differences in parental anxiety. This result contradicts our hypothesis that mothers might have suffered more than fathers due to the pressures of managing the family, maintaining the household, working from home and managing their children’s home schooling. However, previously divergent findings were reported during the pandemic: some studies showed that the female sex was associated with a high level of anxiety [36,37], while others indicated that males and females experienced similar levels of stress, anxiety and negative emotions [3,38,39]. One reason for this finding might be the small number of men in our study. Another explanation for this result in our sample, where there was a majority of two-parent households, might be that fathers who were working from home took on greater childrearing responsibilities and shares of housework, mitigating maternal anxiety levels. Single-parent families, the number of individuals living at home and the number of younger children did not increase the level of parental anxiety in our study. These results contrast with previous findings showing that some family structure characteristics could affect parental well-being [37]; for example, learning conditions for children in middle school and high school could have an effect, with heavier study tasks potentially making parents feel more anxious due to the challenge of self-discipline [3]. According to our results and consistent with previous research [40], one protective factor is high education, which is associated with good psychological functioning.

This research also focused on the impact of living conditions on the psychological status of parents. None of the characteristics of the living space (type and area of residence, access to a private outdoor area) were associated with parental anxiety. A reason for this finding might be the large number of families living in houses with private outdoor access. Our findings revealed that parents with conflicts in the home had significantly higher levels of anxiety. Due to school closures, children and parents had to spend more time together, and children had to study at home. Children’s attitudes and performances that did not meet their parents’ requirements [3] and an increase in recreational screen time [41] might have led to conflicts [3], affecting parents’ anxiety levels. Parenthood is a demanding role, particularly during periods of lockdown, in which parents must take on multiple tasks, increasing the risk of role conflict [42]. In line with previous findings [33,43], the results also showed that having relatives or acquaintances who had been suspected or confirmed to be infected with COVID-19 was associated with anxiety, supporting the direct effect of COVID-19 on anxiety levels in parents who were fearful that they or their family members would become ill and were very uncertain about the repercussions of the pandemic [44].

Some resources can help protect parents from anxiety and promote their well-being. First, during this long period of home confinement and social distancing, parents with higher family support in our study were at a lower risk of experiencing anxiety. Surprisingly, emotional support from family, but not from friends and significant others, was associated with a low level of anxiety. Previous studies have shown that family support, but not support from friends and partners, was associated with low levels of depression and PTSD during the COVID-19 pandemic [45]. In the face of feelings of loneliness and helplessness, friends and significant others could have or could be perceived to have less capacity to validate others’ emotional experiences during this period, while emotional support provided by family might be more meaningful and more stable in providing reassurance. The professional support of some specialists who provided care for children (physicians, psychologists, therapists, etc.) further decreased during the pandemic. The recent literature has underlined the particular need for external specialist support for parents with children with atypical development patterns who experience higher levels of parental distress compared to parents of children exhibiting typical development patterns [46]. This finding is of interest because, in light of the difficulties associated with the global pandemic and the experience of a lack of support from external sources, such as from schools, health professionals or friends, and for some, from family, it is important to support the development of open-access online parenting resources during the COVID-19 pandemic [47]. Second, the role of parental resilience, which is known as a key element of a well-functioning family system and refers to the process of managing stress and functioning well in the face of stressors, adversity, and challenges [48], was confirmed. Lower parental resilience was found to be the main risk factor for anxiety in our study over and above specific COVID-19 variables. In line with previous findings [49,50], this result reflects the importance of boosting individual resilience against psychological distress. In addition, several authors have highlighted the need for nuanced and balanced media news with regard to mental health during the pandemic and resilience, as nocebo effects occur as a result of negative and alarming news coverage [51].

Several limitations should be considered in interpreting the results. First, the study was conducted via an online survey, and the parents of school-aged children who voluntarily completed the survey may have had higher anxiety than those who did not. Thus, our sample results may not be as representative as those from a probability sample and may not allow the generalization of the results. The self-reported data may also lead to response biases for measures where parents provide socially desired results. However, rapid online surveys appeared to be a promising method to assess and track knowledge and perceptions in the midst of rapidly evolving disease outbreaks [52]. Second, due to the study cross-sectional design, the direction of causality cannot be identified. Third, other factors related to participant attrition may have influenced the study results, requiring future investigations, such as studies on other personal (self-report measures of the other parent or partner in the home) and environmental factors (work environment, hours worked at home). In light of these limitations, further longitudinal research is needed in different countries to clarify the relationships. Despite these limitations, this study is strengthened by the use of validated psychological assessment measures and its additional contribution to sociodemographic, environmental and psychological factors connected to higher levels of parental anxiety during the COVID-19 pandemic. Our findings highlight that preventive measures should be directed toward parents who are already known to have a low level of resilience, to prevent amplified negative manifestations that could further impact their children’s mental health.

## 5. Conclusions

The study highlights that more than half of the parents in a French area particularly affected by COVID-19 experienced anxiety during the first wave of the pandemic and suggests some need for planning targeted psychological interventions and support programs for families. The resilience level was a very strong influencing factor of state anxiety during the first wave of the COVID-19 pandemic. Policy makers and mental health professionals should ensure that mental health interventions are available and tailored to help parents cope with public health crises and work-family challenges. From a clinical perspective, this means that counseling psychologists should pay more attention to vulnerable groups, especially those with low levels of resilience, and should develop guidelines for parents experiencing a low level of family support. As parents’ anxiety might increase the likelihood that their children will develop maladaptive behavior, prevention and recommendations are needed in preparation for possible next waves or pandemics.

## Figures and Tables

**Table 1 ijerph-19-10999-t001:** Sociodemographic and working characteristics of the sample population of parents during the lockdown (N = 698).

	Full Sample	
	N	%
**Characteristic**		
**Age** (years)	698	
<35	59	8.5
35–45	361	51.7
>45	278	39.8
**Sex**		
Male	98	14.0
Female	600	86.0
**Marital status** (missing = 3)		
Married/living with the other parent	518	74.5
Separated/divorced/widowed	157	22.6
Single parent	20	2.9
**Living arrangements** (missing = 3)		
Alone with children	110	15.8
With the other parent and children	518	74.5
With a partner other than the parent and children	67	9.6
**Number of individuals confined at home**		
<4	210	30.1
4	318	45.6
>4	170	24.4
**Parental education level** (missing = 1)		
Less than high school	264	37.9
Higher education	433	62.1
**Occupational status before the lockdown** (missing = 11)		
Looking for employment	58	8.4
Full-time work	437	63.6
Part-time work	157	22.9
Retired/student	35	5. 1
**Occupational status during the lockdown** (missing = 31)		
Work interruption	200	30.0
Full-time worker telecommuting from home	243	36.4
Full-time worker at work	149	22.3
Worker working most of the time at work, remaining time at home	32	4.8
Worker working most of the time at home, remaining time at work	43	6.4

**Table 2 ijerph-19-10999-t002:** Living conditions of the sample population of parents during the lockdown (N = 698).

	Full Sample	
	N	%
**Home location** (missing = 9)		
Urban area	271	39.3
Rural area	418	60.7
**Access to a private outside space** (missing = 1)		
No access	38	5.5
Private balcony, courtyard or terrace	79	11.3
Private domestic garden	555	79.6
Courtyard or garden for collective use	25	3.6
**Frequency of exiting the house during the lockdown** (missing = 1)		
Several times a day	101	14.5
Once a day	170	24.4
Several times a week	120	17. 2
Once a week	123	17.6
Less than once a week	91	13.1
Never leaving home	92	13.2
**Difficulty isolating at home**		
Yes	107	15.3
No	591	84.7
**Tensions and conflicts at home**		
Yes	130	18.6
No	568	81.4
**Noises outside the residence**		
Yes	58	8.3
No	640	91.7
**Noises inside the residence**		
Yes	35	5.0
No	663	95.0
**Time spent on children’s schoolwork at home** (missing = 2)		
<2 h a day	300	43.1
2–4 h a day	256	36.8
≥4 h a day	140	20.1
**Someone at home/relative or acquaintance had COVID-19**		
None	403	57.7
Confirmed and hospitalized cases	66	9.5
Confirmed and nonhospitalized cases	144	20.6
Suspected cases	85	12.2

**Table 3 ijerph-19-10999-t003:** Parental anxiety, social support and resilience scores during the lockdown (N = 698).

	Full Sample	
	N	%/Mean (SD)
**GAD-7-total score**	698	5.6 (5.0)
Normal (0–4)	339	48.6
Mild anxiety (5–9)	225	32.2
Moderate anxiety (10–14)	91	13.0
Severe anxiety (15–21)	43	6.2
**SF-12 PCS score**	698	69.6 (13.0)
**SF-12 MCS score**	698	55.4 (16.9)
**MSPSS-total score**	698	5.5 (1.2)
**MSPSS-subscales**		
Family	698	5.5 (1.3)
Friends	698	5.3 (1.4)
Significant other	698	5.7 (1.3)
**BRS-total score**	698	3.5 (0.8)

Note: SD, standard deviation; GAD-7, 7-item Generalized Anxiety Disorder Scale; MSPSS, Multidimensional Scale of Perceived Social Support; BRS, Brief Resilience Scale.

**Table 4 ijerph-19-10999-t004:** Factors associated with moderate to severe parental anxiety levels during the COVID-19 lockdown (N = 698).

	Bivariate Regression Analysis			Multivariate Logistic Regression Analysis R^2^ = 0.3, H&L = 0.83		
	OR	95% CI	*p* Value	OR	95% CI	*p* Value
**Age** (ref: ≥45 years)			0.076			
<35	2.0	1.0–3.7				
35–44	1.0	0.7–1.5				
**Living arrangements** (ref: with the other parent or partner and children)			0.107			
Alone with children	1.5	0.9–2.4				
**Parental education level** (ref: higher education)			<0.001			0.001
Less than high school/High school graduate	2. 1	1.4–3.1		2.1	1.3–3.2	
**Time spent on children’s schoolwork at home** (ref: <2 h a day)			0.047			
2–4 h a day	1.1	0.7–1.7				
≥4 h a day	1.8	1.1–2.9				
**Home location** (ref: urban area vs. rural area)	1.0	1.1–2.9	0.856			
**Occupational status during the lockdown** (ref: work interruption)			0. 227			
Full-time worker telecommuting from home	0.6	0.4–1.0				
Full-time worker at work	0.9	0.6–1.6				
Worker working most of the time at work, remaining time at home	0.5	0. 2–1.5				
Worker working most of the time at home, remaining time at work	0.6	0. 2–1.5				
**Number of individuals confined at home** (ref: <4)			0.551			
4	0.8	0.5–1.2				
>4	0.9	0.5–1.5				
**Access to a private outside space** (ref: no access)			0. 218			
Private balcony, courtyard or terrace	1.0	0.4–2.8				
Courtyard or garden for collective use	2.5	0.8–7.9				
Private domestic garden	1.0	0.4-.2.4				
**Difficulty isolating at home** (Yes vs. No)	2.4	1.5–3.8	<0.001			
**Tension and conflicts at home** (Yes vs. No)	3.0	2.0–4.6	<0.001	2.3	1.4–3.7	0.001
**Noises outside the residence** (Yes vs. No)	2.6	1.5–4.7	<0.001	2.0	1.0–3.9	0.046
**Noises inside the residence** (Yes vs. No)	3.0	1.5–6.1	<0.001			
**Someone at home/relative or acquaintance infected with COVID-19** (ref: no)			0.001			<0.001
Confirmed and hospitalized cases	3.1	1.7–5.4		3.8	2.0–7.3	
Confirmed and nonhospitalized cases	1.6	1.0–2.6		1.8	1.0–3.1	
Suspected cases	1.6	0.9–2.9		1.9	1.0–3.8	
**MSPSS-subscales**						
Family	0.6	0.5–0.7	<0.001	0.7	0.6–0.8	<0.001
Friends	0.7	0.6–0.8	<0.001			
Significant other	0.7	0.6–0.8	<0.001			
PCS SF-12 score (ref: score ≥ the median PCS score)	1.8	1. 2–2.7	0.003			
BRS-total score (ref: score ≥ 3)						
BRS score < 3	5. 9	3.9–8.9	<0.001	4.3	2.7–6.7	<0.001

Note: OR, odds ratio: the probability of GAD-7 scores > 10; OR < 1, decreased frequency of GAD-7 scores > 10; OR > 1, increased frequency of GAD-7 scores > 10; SD, standard deviation; GAD-7, 7-item Generalized Anxiety Disorder Scale; MSPSS, Multidimensional Scale of Perceived Social Support; BRS, Brief Resilience Scale.

## Data Availability

The data collected and analyzed during the current study are available from the corresponding author upon request.

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
