# Peer review of "Mental Health Status of French School-Aged Children’s Parents during the COVID-19 Lockdown and Its Associated Factors"

_ijerph, 2022, doi:10.3390/ijerph191710999_

Round 1

Reviewer 1 Report

The research presented aims to analyze the levels of anxiety suffered by parents of school-age children during the COVID-19 pandemic. The study is interesting and can be of great use in helping families psychologically, especially considering that we still have the COVID problem, although it is now milder. It is recommended that the authors specify the number of participants in the section on participants and procedure, since this is only reflected in the results. In general, the research is good and clearly presented.

Reviewer 2 Report

Dear Authors:

It is recommended to replace "epidemic" with "pandemic".

In the Introduction, the authors introduced the background and significance of the research, research status, and research gaps. I think the descriptions in this section are substantial and easy to understand. However, a description of the structure of the paper should be added.

The second section of the paper was supposed to be a literature review, but it was missing from the author's paper. In this section, the relationship among the various variables should be discussed in the literature review and relevant hypotheses should be proposed, as well as the construction of theoretical models.

Based on my last suggestion, after the theoretical model is proposed, the data analysis method should also be adjusted, such as using Amos or Lisrel. Rather than just showing the variable's scores of means and standard deviations, such a simple analysis is obviously not enough for the depth of the study.

The authors did not discuss the relationship among variables, even for cross-sectional studies the relationship among variables could be discussed. I highly recommend that the authors redesign the entire study from the second section, otherwise, I do not consider the paper acceptable.

Reviewer 3 Report

The study Mental health status of French school-aged children’s parents during the COVID-19 lockdown and its associated factors provides a detailed exploration of the factors that influence parents’ mental health status during the COVID-19 lockdown, and is of great significance and importance to help parents and families during this particular period. Some comments are provided as follows.

1.       No description of the participants was found. Please include details.

2.       Could you please provide children’s Mage and SD?

3.       How did you deal with siblings? Did parents fill in more than one questionnaire?

4.       The validity of the measurement instruments used in this study needs to be provided. Please include the Cronbach's alphas for all the measures.

5.       The Cronbach's alpha for the GAD-7 (line 110) was the validity of the cited reference (ref. 22). Moreover, it would be helpful to provide some sample items for understanding of the measured variables.

6.       The reference in line 112 has validated the French version of the GAD-7 in a sample of people with epilepsy. Here, it is suggested to cite the French version of the GAD-7 which is suitable for the general population.

7.       Did the sociodemographic data and other characteristics come from a validated tool? If not, please provide info about the psychometrics of the scale.

8.       From the results, Table1 (line 190) and Table 2 (line 192) do not need to write “mean (SD)”, as the following results are all percentages.

9.       The formatting of the table needs to be checked again, for example line 192 showed "Full sample N=698", but line 190 did not. In general, tables do not need to specifically show "Abbreviation", but can simply use "Note".

10.    Lines 153-155 mentioned that the outcome was a GAD-7 score higher than 10, so has the authors tried to separately explore moderate and high levels of anxiety? Why were parents with low anxiety not considered here? Perhaps it would be interesting to find the factors associated with low anxiety, and it might be possible to find some protective factors.

11.    It is suggested that the p value of Table 4 could report two or three decimals.

12.    Due to the specificity of the lockdown during the COVID-19, has the authors modified the content of the instruments to make them more appropriate for use in this particular situation? For example, the MSPSS, because of the lockdown, there are some supports (e.g., behavioral supports) might not be provided. So, how do we consider this phase of the supports?

13.    Line 94 mentioned that the data were collected under a partial lockdown. Could the authors provide more information about this situation? Was it part-area lockdown, part-time lockdown, or something else?

14.    Why not include MCS into the regression analysis?

15.    Could you provide the results of the multicollinearity test to ensure that there was no collinearity issue among the variables? For example the VIF value.

16.    Lines 235-245: similarities/differences across studies are not clear. Please work on it.

17.    The format of the references needs to be checked again.

Reviewer 4 Report

Thank you very much for offering me the opportunity to review this  manuscript. I think that the manuscript could be an interesting paper. However, in my opinion, it presents some problems.

Participants

Methods:

Participants: sample selection should be described in more detail by reporting e.g. inclusion/exclusion criteria and some data described in the “Sociodemographic and living condition characteristics” section.

INSTRUMENTS

This section presents many concerns. Please, add some examples of items for each measure

CONCLUSIONS

I suggest to enrich this section with psychological intervention addressed to families (see:  Polizzi, C., Burgio, S., Lavanco, G., & Alesi, M. (2021). Parental Distress and perception of children’s executive functioning after the first COVID-19 lockdown in Italy. Journal of Clinical Medicine10(18), 4170.)

The authors should mention the added value of this work in relation to other published articles

English needs addressing

Round 2

Reviewer 2 Report

Thanks for the edit.

Reviewer 3 Report

Thank you for the opportunity to revise the current manuscript. The authors did a good job and addressed all the previous comments. I suggest to consider it for publishing.